# The Antibodies’ Response to SARS-CoV-2 Vaccination: 1-Year Follow Up

**DOI:** 10.3390/biomedicines11102661

**Published:** 2023-09-28

**Authors:** Eleonora Nicolai, Flaminia Tomassetti, Martina Pelagalli, Serena Sarubbi, Marilena Minieri, Alberto Nisini, Marzia Nuccetelli, Marco Ciotti, Massimo Pieri, Sergio Bernardini

**Affiliations:** 1Department of Experimental Medicine, University of Rome Tor Vergata, Via Montpellier 1, 00133 Rome, Italy; nicolai@med.uniroma2.it (E.N.); flaminia.tomassetti@students.uniroma2.eu (F.T.); martina90c@hotmail.it (M.P.); serena.sarubbi@gmail.com (S.S.); minieri@med.uniroma2.it (M.M.); bernards@uniroma2.it (S.B.); 2Department of Laboratory Medicine, Tor Vergata University Hospital, Viale Oxford 81, 00133 Rome, Italy; marzianuccetelli@yahoo.com; 3Department of Diagnostic Imaging and Radiology, Tor Vergata University Hospital, Viale Oxford 81, 00133 Rome, Italy; alberto.nisini@ptvonline.it; 4Department of Laboratory Medicine, Virology Unit, Tor Vergata University Hospital, Viale Oxford 81, 00133 Rome, Italy; marco.ciotti@ptvonline.it

**Keywords:** SARS-CoV-2, antibodies, serological test, vaccine

## Abstract

The use of vaccines has allowed the containment of coronavirus disease 2019 (COVID-19) at a global level. The present work aims to add data on vaccination by evaluating the level of neutralizing antibodies in individuals who have received a three-vaccination series. For this purpose, we ran a surveillance program directed at measuring the level of IgG Abs against the Receptor Binding Domain (RBD) and surrogate virus neutralizing Ab (sVNT) anti-SARS-CoV-2 in the serum of individuals undergoing vaccination. This study was performed on employees from the University of Rome Tor Vergata and healthcare workers from the University Hospital who received the Vaxzevria vaccine (n = 56) and Comirnaty vaccine (n = 113), respectively. After the second dose, an increase in both RBD and sVNT Ab values was registered. In individuals who received the Comirnaty vaccine, the antibody titer was about one order of magnitude higher after 6 months from the first dose. All participants in this study received the Comirnaty vaccine as the third dose, which boosted the antibody response. Five months after the third dose, nearly one year from the first injection, the antibody level was >1000 BAU/mL (binding antibody units/mL). According to the values reported in the literature conferring protection against SARS-CoV-2 infection, our data indicate that individuals undergoing three vaccine doses present a low risk of infection.

## 1. Introduction

The spread of SARS-CoV-2 infection has forced the whole world to face this new emergency. Millions of people have been affected globally by coronavirus disease 2019 (COVID-19), since the World Health Organization (WHO) declared it a pandemic on 11 March 2020 (Report WHO). To bring an end to this pandemic, which has resulted in severe clinical, economic, and social repercussions, concerted efforts worldwide were required to develop vaccines and make them available [1]. Indeed, the urgent need for safe and effective prophylactic vaccines was crucial in achieving herd immunity.

Since the discovery of SARS-CoV-2 virus and its genome characterization, more than 300 vaccine projects [2] have been developed by the scientific community, and over 40 have now been approved by authorities (BC CENTRE). On 14 December 2020, the first COVID-19 vaccination had taken place in the United States, and many countries had been gearing up for the largest-ever immunization campaigns (WHO; ECDC). The first vaccine to receive emergency approval from the US Food and Drug Administration (FDA) and the European Medicine Agency (EMA) was BNT162b known as Comirnaty (Pfizer/BioNTech) [3,4]. Comirnaty is an mRNA vaccine that encodes the full-length Spike protein of SARS-CoV-2 and is administered in two doses, spaced 21 days apart for the initial vaccination regimen. It has demonstrated 95% protection against COVID-19 in individuals aged 12 and older, achieving robust immunity [5]. Several studies consistently reported that the vaccine generates an immune response against RBD (receptor binding domain) within the S1 Spike protein [6,7]. Furthermore, a second dose enhances the immune response, resulting in an increased level of binding and neutralizing antibodies [8,9]. Similarly, for AstraZeneca’s Vaxzevria (AZD1222), the level of induced antibodies is low after the first dose but significantly improves after the second, providing enhanced immune protection [10]. Vaxzevria is an adenovirus-based vaccine that is replication-deficient and has been modified to contain the gene encoding the Spike protein of SARS-CoV-2 [11].

On 1 January 2021, the Italian health authorities commenced the distribution of the Comirnaty vaccine, followed by the Vaxzevria vaccine. By the beginning of 2022, Comirnaty had also received approval for use in children, and in April 2022, 80% of the population had been vaccinated against COVID-19 [12]. Post-vaccination neutralizing antibody titers, measured through serological assays, exhibit a strong correlation with protection from SARS-CoV-2 disease, although this relationship has been widely discussed in the literature [13,14]. Additionally, both humoral- and cell-mediated immune responses are associated with vaccine-induced protection [7]. Nonetheless, further studies are necessary to enhance our understanding of the immunological response elicited by the vaccine. While specific high antibody titers can block the infecting virus, a combination of humoral and cellular immunity is essential for virological control following a breakthrough infection [15]. It would be important to improve our understanding of the role of mucosal humoral and cellular immunity at the point of entry, which might have an important role in protecting against SARS-CoV-2 infection.

Recent studies have suggested that a single dose of an mRNA vaccine elicits a very rapid immune response, comparable with the response in individuals with prior SARS-CoV-2 exposure [16,17]. Concurrently, other studies have demonstrated that neutralizing activity significantly increases after two of mRNA vaccine doses, suggesting that infection induces a robust B-cell memory response, resulting in similar neutralizing antibody levels of those observed in pre-infected individuals [18,19].

However, data on vaccine efficacy over time remain limited, despite their significance in limiting the spread of SARS-CoV-2. For instance, it has been demonstrated that the titer of post-vaccination neutralizing antibodies (Abs) correlate with protection from SARS-CoV-2 infection [16]. This study aims to enhance the available vaccination data by evaluating the titer of surrogate virus neutralizing antibodies in individuals who received a triple-dose regimen. A comprehensive assessment is needed to evaluate the post-vaccination antibody trend over several collection times [20].

Recent studies have suggested an Abs’ cut-off value of immunity against SARS-CoV-2 infection ranging from 200 to 800 BAU/mL [21,22,23]. Based on such values, our data indicate that individuals who underwent a three-vaccination series present a low risk of infection one year after vaccination.

## 2. Materials and Methods

### 2.1. Sample Collection

One-hundred-sixty-nine healthy individuals (aged between 25 and 65 years old) were monitored during their vaccination program: 113 healthcare workers of Tor Vergata COVID-19 Hospital received the Comirnaty vaccine, and 56 workers of the University of Tor Vergata received the Vaxzevria vaccine, from January 2021 to February 2022. University employees received two doses of Vaxzevria vaccine 12 weeks apart, while hospital workers received two doses of Comirnaty 3 weeks apart. All patients received the Comirnaty vaccine as third dose. Collection of serum samples occurred between the first and second vaccine dose, between the second and third dose, and then after the third dose. All subjects underwent periodical blood draw collection for measuring the level of antibodies against SARS-CoV-2. In the Vaxzevria group, serum was obtained at about 21, 35, 50, 80, and 110 days after the first vaccine dose. In the Comirnaty group, serum was collected 21 days after the first dose (before the second dose injection); about 35, 50, 80, and 110 days after the second dose; and about 30 days from the third dose for both groups. All participants in this study were monitored for SARS-CoV-2 infection through nasopharyngeal swab, by the surveillance protocol. The positive subjects were excluded from this study.

This study was carried out in accordance with the Declaration of Helsinki and approved by the local Ethical Committee of the Tor Vergata University Hospital of Rome (protocol no. R.S.44.20). All subjects enrolled in this study gave informed consent.

### 2.2. Determination of IgG Antibodies against SARS-CoV-2 S-RBD by SNIBE

The Maglumi SARS-CoV-2 S-RBD is an indirect chemiluminescence immunoassay (CLIA) for the semi-quantitative determination in vitro of IgG antibodies against SARS-CoV-2 RBD sequence within the S protein, run on the fully automated Maglumi 800 analytical system (Snibe Diagnostic, Shenzhen, China).

The serum sample was thoroughly mixed with magnetic beads coated with RBD recombinant antigen and buffer solution. After placing the obtained mixture in a magnetic field, the supernatant was removed and a wash cycle was performed. At this point, anti-human IgG antibodies labeled with amino-butyl-ethyl-isoluminol (ABEI) were added and incubated to form immune complexes. A second precipitation in a magnetic field was performed, followed by removal of the supernatant and a new wash cycle. Then, a starter reagent was added and a chemiluminescent reaction was initiated, which produced a light signal measured using a photomultiplier as relative light units (RLUs), which was proportional to the concentration of anti-SARS-CoV-2 RBD IgG present in the sample. According to the recommendations of the World Health Organization (WHO) International Standard (IS) to harmonize the measurement units for binding activity in antibody assays, the BAU/mL (Binding Antibody Units) measurement unit is adopted [24].

Samples with values > 100 AU/mL were diluted 1:10/1:20 with an extension of the dynamic range to 2000 AU/mL.

The manufacturer declares a cut-off value > 1 AU/mL and an intra- and inter-assay precision between 1% and 4%, with a clinical specificity of 99.6% (95% CI: 98.7–100%) and a cumulative sensitivity of 100% (95% CI: 99.9–100%), calculated at 15 days or more after the first positive PCR.

### 2.3. Determination of IgG/IgA/IgM Surrogate Virus Neutralizing Antibodies against SARS-CoV-2 by SNIBE

The automated Maglumi analytical system (SNIBE Diagnostics, Shenzhen, China) allows the detection of anti-SARS-CoV-2 IgG/IgA/IgM surrogate virus neutralizing antibodies (sVNT) using a competitive chemiluminescent immunoassay (CLIA). The assay mimics the virus–host interaction using the highly specific ACE2-RBD protein interaction. In the presence of surrogate virus neutralizing antibodies, the binding between the RBD antigens and ACE2 is prevented because of the competition of surrogate virus neutralizing antibodies with RBD antigens.

Magnetic beads coated with specific ACE2 antigens and SARS-CoV-2 RBD recombinant antigens labeled with amino-butyl-ethyl-isoluminol (ABEI) were mixed thoroughly with sample and buffer solution, and incubated. Anti-SARS-CoV-2 sVNT antibodies present in the serum sample competed with ACE2 antigen immobilized on magnetic microbeads for the binding to recombinant SARS-CoV-2-RBD-specific antigen labeled with ABEI. Following precipitation in a magnetic field, the supernatant was removed and a washing step was performed. Then, starter reagents were added to initiate a chemiluminescent reaction, which generated a light signal measured using a photomultiplier as relative light units (RLUs), which was inversely proportional to the concentration of SARS-CoV-2 neutralizing antibodies present in the sample.

According to the manufacturer, the cut-off value is 0.03 µg/mL and the intra- and inter-assay precision is between 1.27% and 1.01%, with a clinical specificity of 100% (95% CI: 93.69–100.00%) and a cumulative sensitivity of 100% (95% CI: 96.90–100.00%), calculated at 14 days or more, after the first positive PCR.

### 2.4. Determination of IgG Antibodies against SARS-CoV-2 S-RBD by Mindray

The Mindray SARS-CoV-2 S-RBD IgG (Mindray S-RBD IgG) is a quantitative CLIA assay performed in two steps on the fully automated Mindray CL 1200i analytical system (Mindray Bio-Medical Electronics Co., Ltd., Shenzhen, China). In the first step, serum sample, sample treatment solution, and paramagnetic microparticles coated with SARS-CoV-2 RBD antigen were dispensed into a reaction vessel and incubated. The SARS-CoV-2 RBD IgG antibodies present in the sample bound to RBD antigen coated on the microparticles. These complexes were magnetically captured while unbound substances were removed through washing. In the second step, alkaline phosphatase (ALP)-labeled anti-human IgG monoclonal antibodies and diluent solution were added to the reaction vessel to form a sandwich structure with the microparticle-captured RBD IgG antibodies. Next, a substrate solution was added followed by a chemiluminescent reaction measured as RLUs using a photomultiplier. The quantity of anti-RBD IgG antibodies present in the sample was proportional to the RLUs generated during the reaction. According to the manufacturer, the cut-off value in AU/mL is 10.0, while that in BAU/mL is 12.16, and the linearity was demonstrated in the range of 3 AU/mL to 1000 AU/mL. Samples with values > 1000 AU/mL were diluted at 1:10, allowing the extension of the dynamic range of the assay to 10,000 AU/mL.

According to the manufacturer, the intra- and inter-assay precision is between 1.7% and 4.08%, with a clinical specificity of 99.6% (95% CI: 99.3–99.8%) and a cumulative sensitivity of 99.6% (95% CI 98.9–99.8%), calculated at 14 days or more, after the first positive PCR.

### 2.5. Determination of IgG antibodies against SARS-CoV-2 sVNT by Mindray

The Mindray SARS-CoV-2 neutralizing antibody assay is a chemiluminescent immunoassay used for the quantitative determination of SARS-CoV-2 surrogate virus neutralizing antibodies in human serum or plasma. The surrogate neutralizing antibodies block the interaction between the RBD of the viral spike glycoprotein and the ACE2 cell surface receptor.

As the first step, the patient’s sample, sample treatment solution, paramagnetic microparticles coated with SARS-CoV-2 antigens, and ACE2-alkaline phosphatase (ALP) conjugate were dispensed into a reaction cuvette and incubated. Following incubation, the neutralizing antibodies present in the sample competed with the ACE2-ALP conjugate for binding sites of SARS-CoV-2 antigens. The neutralizing antibody or ACE2-ALP conjugate bound to the antigen on the microparticles, which were magnetically captured, while unbound substances were removed through washing. In the second step, a substrate solution was added to the reaction cuvette. A chemiluminescent reaction occurred, which was measured as relative light units (RLUs) using a photomultiplier built inside the system. The level of neutralizing antibodies present in the sample was inversely proportional to the relative light units (RLUs) generated during the reaction.

According to the manufacturer, the cut-off value is 1 IU/mL and the intra- and inter-assay precision is between 1.7% and 4.08%, with a clinical specificity of 99.6% (95% CI: 99.3–99.8%) and a cumulative sensitivity of 99.6% (95% CI 98.9–99.8%), calculated at 14 days or more, after the first positive PCR.

## 3. Results

Over the past two years, we have conducted a surveillance program involving individuals who received the Vaxzevria/AstraZeneca and Comirnaty/Pfizer-BioNTech vaccines. The primary objective of this program was to quantify the level of IgG antibodies against the RBD of the viral spike glycoprotein, as well as sVNT anti-SARS-CoV-2, which inhibits the interaction between RBD and the ACE2 receptor. The antibody concentrations were assessed in the serum of individuals after the first, the second, and the third dose of the vaccine.

A total of 2522 Ab measurements were conducted, employing two different automated platforms.

Table 1 provides information regarding the age and gender of the individuals included in this program.

Both vaccines were initially administered in two doses: Vaxzevria with a 12-week interval and Comirnaty with a 3-week interval. Ab titers were measured using two independent methods. Figure 1, Figure 2 and Figure 3 display the values of RBD and sVNT Abs measured after each vaccine dose over time, starting from the administration of the first dose.

Following the second dose, both vaccines exhibited an increase in anti-RBD and sVNT antibody levels. Due to the differing time intervals between the first and second dose, the Comirnaty vaccine yielded higher Abs concentrations earlier than Vaxzevria did. Moreover, the Ab levels measured in the serum of subjects who received the Comirnaty vaccine were consistently higher by up to an order of magnitude. Five months after the first dose (0 ÷ 150 days), the average Ab titer was 1260 BAU/mL for Comirnaty and 157 BAU/mL for Vaxzevria (values registered using Mindray platform).

In November 2021, the Italian government decided to administer a third dose. All patients received the Comirnaty vaccine, regardless of the previous ones. This booster dose led again to an increase in Ab levels. Individuals who had previously received the Comirnaty vaccine recorded values equal to or higher than those observed after the second dose. In contrast, subjects who had initially received the Vaxzevria vaccine exhibited values much higher than those observed after the second dose, reaching levels comparable to individuals vaccinated with Comirnaty. Within the 150 ÷ 350 day range, the average Ab titer was 2400 BAU/mL for Comirnaty and 1650 BAU/mL for Vaxzevria, as shown in Figure 1.

Antibodies levels were also assessed using the SNIBE SARS-CoV-2 S-RBD kit and anti-SARS-CoV-2 IgG/IgA/IgM Neutralizing Antibodies (NTs). This analysis yielded a similar trend, as illustrated in Figure 3. For the sake of clarity, median and quartile data are summarized in Table 2.

To highlight the interpersonal variability in antibody production, the time course of the antibodies for five randomly selected individuals among the study participants was analyzed and is reported in Figure 4.

For both vaccines, the Ab trend as a function of time follows those shown in the previous figures. It can be noted that the third dose on individuals who previously received the Vaxzevria vaccine gives a booster effect from the point of view of the Ab concentration, both for RBD and sVNT. However, it is noted that for both vaccines, there is an individual component that determines the extent of the increase in the Ab level.

Therefore, considering an antibody concentration between 200 and 800 BAU/mL as protective, the figures highlight that after the two doses of Comirnaty and already after one month from the first immunization, people are protected. On the contrary, people initially vaccinated with Vaxzevria do not show concentrations of protective antibodies until 240 days, after the booster dose with the mRNA vaccine.

This indicates that the immune response to a vaccine varies from person to person; indeed, several factors such as age, health status, and the presence of pre-existing diseases may affect vaccine effectiveness [25].

However, although Ab titer does not directly measure the level of protection, the results suggest that the third dose of vaccines can reduce the risk of infection.

## 4. Discussion

The waning of immunity induced by the vaccine has been attributed to the decline in anti-S antibodies and the emergence of variants. Anti-S antibody levels begin to decrease 2–3 months after vaccination and continue up to 8 months post vaccination [26,27,28]. A similar trend has been observed for anti-S and anti-N antibodies after natural infection [29,30]. Antibody decay is a natural phenomenon to prevent the persistence of circulating immunoproteins (with a half-life of 50–60 days). However, the reduction in serum antibody levels does not imply a complete loss of immune protection [31]. Indeed, another aspect that should be considered in immunological monitoring is the cellular response (T cell) against SARS-CoV-2, which could complement the humoral response [32,33]. Currently, emerging tests for the detection of T cells should be included into clinical laboratory diagnostics to provide a comprehensive assessment of individual protection. Furthermore, it could also be interesting to conduct a comparative evaluation of the RBD and sVNT IgGs and secretory IgAs and mucosal immunity. Presently, contrasting data have been reported in previous studies regarding COVID-19 vaccines and their ability to elicit a durable IgA response [34,35].

In this study, we assessed the levels of RBD and sVNT IgGs in individuals who received a three-vaccination series, observed over one year. Our results showed that the immune response to two doses of the Vaxzevria vaccine initially lags significantly behind the response induced by Comirnaty. This finding is consistent with numerous studies that have reported a mere 30% increase in antibody levels following the second dose of viral vector vaccines such as Vaxzevria/AstraZeneca, in contrast to a 300–400% increase seen with mRNA vaccines like Comirnaty/Pfizer [17,36,37]. The data also demonstrate that following the administration of a third dose to individuals previously vaccinated with Vaxzevri, Ab levels rise to values comparable to those achieved with homologous vaccination using the mRNA vaccine, as similarly observed in other studies [17,38]. This trend is well depicted in Figure 4, where it can be observed that the second vaccine doses result in higher Ab levels, which are further enhanced after the heterologous third dose, the booster dose, leading to a more robust immune response. This suggests the possibility of heightened protective efficacy, in line with recent research findings [9]. However, the Ab levels exhibit a slight decline up to 6 months post vaccination, consistent with previous reports [39]. The natural decay, as observed for other vaccine-induced Abs, is not a symptom of decreased immune protection given the presence of memory B cells that rapidly expand and differentiate into Ab-secreting plasma cells upon re-exposure [40].

The RBD of the S antigen is the target of most neutralizing antibodies (NTs). Indeed, RBD mediates viral entry by binding the ACE2 receptor present on the surface of human cells. Studies showed that the level of virus NT Abs correlates with protective immunity [41,42,43]; furthermore, although the virus neutralization test (NT) still remains the gold standard, sVNT Abs also demonstrated a good agreement with NT Abs (equal to 98.86%) [44]. It should also be considered that the diagnostic techniques used for Ab detection in clinical laboratories are based on Ab against the ancestral antigen [45,46]. Viral variants such as the Omicron variant are characterized by numerous mutations in the RBD, making this type of monitoring unreliable and not reflecting the individual’s real state of protection [45,47,48]. For example, it has been reported that, today, vaccines are not very effective against infection and transmission of the Omicron variant, even at the peak of the immune response after boosting. Thus, it is likely that the level of neutralizing antibodies needed to protect against infection with the highly transmissible Omicron variant should be much higher than that required to protect against previous variants [15,49].

An important aspect to consider in the administration of vaccine doses is the individual’s humoral response, which is strictly personal and influenced by natural or pathological immunosuppression, as well as external factors, such as drugs administered, senility, smoking, and obesity. All of these factors may lead to a vaccine non-response [49]. It is noteworthy that individuals who have previously experienced COVID-19 may exhibit a strong immune response following the first dose of the vaccine [50,51,52], highlighting the subjectivity of the immune response.

Therefore, when studying COVID-19 vaccines and booster shots, it is important to assess not only short-term neutralizing antibody titers but also the persistence of antibody responses, memory B-cell responses, and cross-reactive T-cell responses [15]. Additionally, it would be beneficial to incorporate diagnostic methods for measuring secretory IgAs levels in biological samples, such as saliva. IgA antibodies play a significant role in the immune defense of mucosal surfaces. They are likely the most important immunoglobulins to fight pathogens in the respiratory and digestive systems at the site of entry of the microorganism. Specifically, secretory IgAs released into the mucus can neutralize SARS-CoV-2 before it reaches and binds epithelial cells. Consequently, the development of mucosal vaccines administered orally or nasally, targeting the RBD of SARS-CoV-2 and inducing the production of secretory IgAs, might neutralize the virus at its point of entry, thereby preventing COVID-19.

## 5. Conclusions

Vaccines have played a pivotal role in combating the COVID-19 pandemic by conferring protection against severe infection. IgG levels have been utilized as a surrogate measure for neutralizing antibody titers and protection. Therefore, comprehending the long-term kinetics of antibodies is pertinent for estimating individual protection and curtailing the spread of COVID-19. Currently, only a limited number of studies have reported data on the IgG anti-spike (RBD) antibody response extending beyond 6 months post-SARS-CoV-2 vaccination. Our study aims to augment the existing literature on this topic by providing data up to 1 year following the first vaccine dose. This information will be instrumental in identifying individuals at heightened risk of infection who may necessitate a booster dose. Thus, confirming and emphasizing the significant value of serological tests assumes particular significance in the management of a vaccine campaign. Further research is needed to optimize the global vaccination program, especially considering that the virus has now become endemic in the population. In conclusion, vaccination against the original strain confers protective antibody titers against infection; however, the risk of infection is greater with Omicron sublineages, even though vaccination protects against severe outcomes due to conserved T-cell immunity. As SARS-CoV-2 is expected to undergo ongoing evolution, the development of variant-adapted vaccines is imperative to extend the duration and breadth of protection against infection. Understanding a voluntary-based screening program, as the one described here, can offer insights into future public health strategies and campaigns. Moreover, as COVID-19 vaccines can confer long-lasting humoral immunity, comprehensive knowledge of the immune response is crucial for developing personalized vaccine and treatment plans. This becomes especially important when considering global vaccine accessibility and prioritizing the administration of booster doses to individuals most at risk.

## Figures and Tables

**Figure 1 biomedicines-11-02661-f001:**
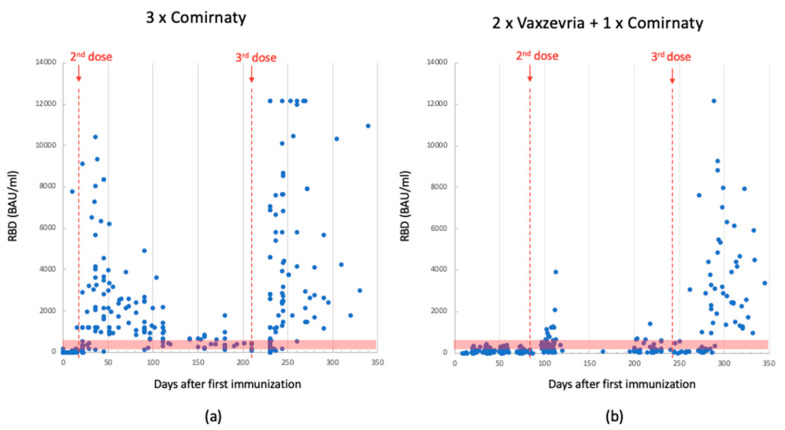
Anti-RBD Ab levels detected using Mindray in 3-dose-Comirnaty-vaccinated individuals; data collected between the 1st and 2nd dose (median: 323; quartile (1st–3rd): 30–1504) and after the 3rd dose (median: 378; quartile (1st–3rd): 152–2592) (panel (**a**)). Anti-RBD Ab levels detected using Mindray in 2-dose-Vaxzevria- plus 1-dose-Comirnaty-vaccinated individuals; data collected between the 1st and 2nd dose (median: 74; quartile range (1st–3rd): 38–159) and after the 3rd dose (median: 282; quartile (1st–3rd): 102–2436) (panel (**b**)). The red dotted lines indicate the time of dose injection. The red band indicates the antibody concentration between 200 and 800 BAU/mL, which is the range reported in the literature considered protective against severe SARS-CoV-2 infection.

**Figure 2 biomedicines-11-02661-f002:**
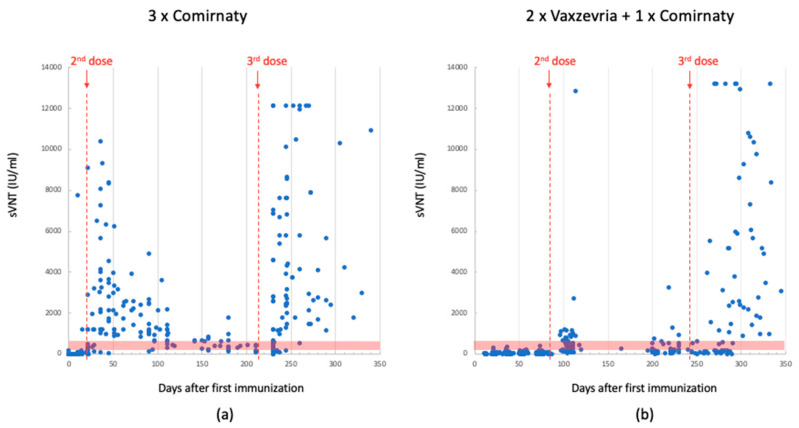
sVNT Ab levels detected using Mindray in 3-dose-Comirnaty-vaccinated individuals. Data between the 1st and 2nd dose (median: 126; quartile (1st–3rd): 6–940); data after the 3rd dose: (median: 465; quartile (1st–3rd): 189–4716) (panel (**a**)). SVNT Ab levels detected using Mindray in 2-dose-Vaxzevria- plus 1-dose Comirnaty-vaccinated individuals. Data between the 1st and 2nd dose (median: 14; quartile (1st–3rd): 6–35); data after the 3rd dose (median: 151; quartile (1st–3rd): 32–933) (panel (**b**)). The red dotted lines indicate the time of dose injection. The red band indicates the antibody concentration between 200 and 800 BAU/mL, which is the range reported in the literature considered protective against severe SARS-CoV-2 infection.

**Figure 3 biomedicines-11-02661-f003:**
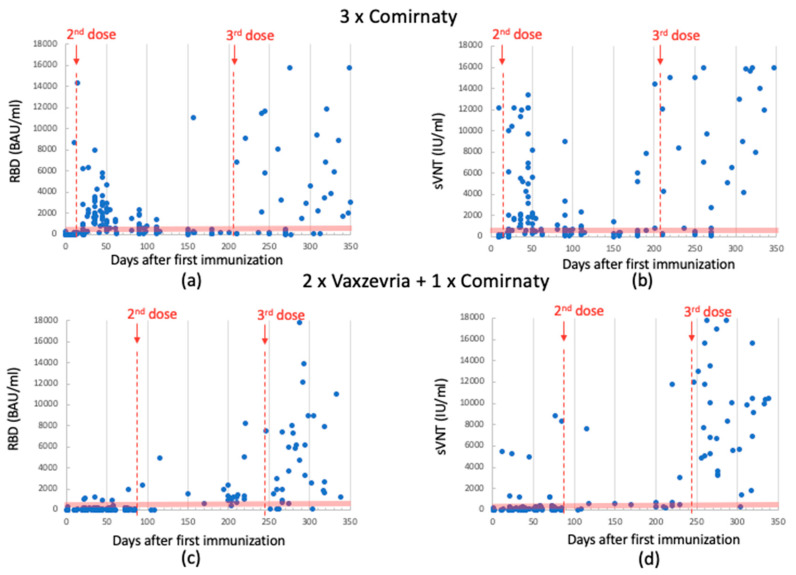
Anti-RBD Ab levels detected using Snibe in 3-dose-Comirnaty-vaccinated individuals. Data between 1st and 2nd dose (median: 359; quartile (1st–3rd): 45–1461); data after 3rd dose (median: 219; quartile (1st–3rd): 89–4253) (panel (**a**)). sVNT Ab levels detected using Snibe in 3-dose-Comirnaty-vaccinated individuals. Data between 1st and 2nd dose (median: 659; quartile (1st–3rd): 211–2058); data of 3rd dose (median: 788; quartile (1st–3rd): 261–9667) (panel (**b**)). Anti-RBD Ab levels detected using SNIBE in 2-dose-Vaxzevria- plus 1-dose-Comirnaty-vaccinated individuals. Data between 1st and 2nd dose (median: 47; quartile (1st–3rd): 24–93)); data after 3rd dose (median: 1976; quartile (1st–3rd): 1062–6188) (panel (**c**)). sVNT Ab levels detected using SNIBE in 2-dose-Vaxzevria- plus 1-dose-Comirnaty-vaccinated individuals. Data between 1st and 2nd dose (median: 96; quartile (1st–3rd): 59–160); data after 3rd dose (median: 7784; quartile (1st–3rd): 3100–11,850) (panel (**d**)). The red dotted lines indicate the time of dose injection. The red band indicates the antibody concentration between 200 and 800 BAU/mL, which is the range reported in the literature considered protective against severe SARS-CoV-2 infection.

**Figure 4 biomedicines-11-02661-f004:**
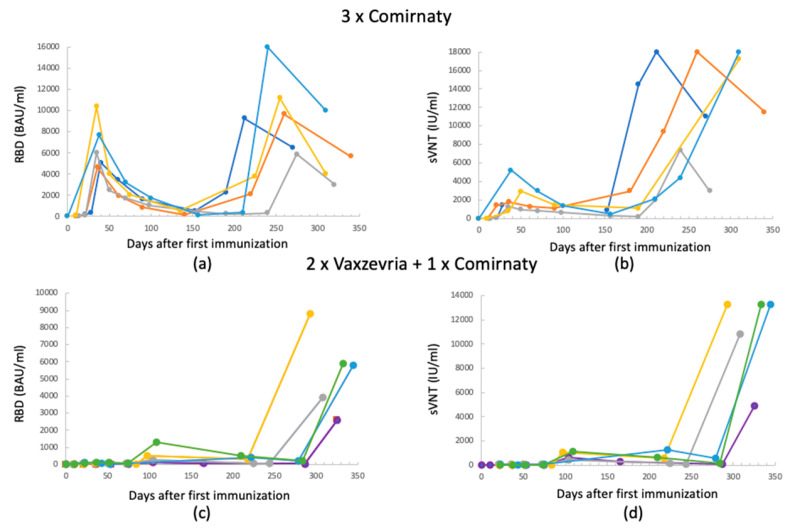
Antibodies’ level time course for 5 randomly selected persons: (**a**,**b**) 3 × Comirnaty RBD and sVNT Abs, respectively; (**c**,**d**) 2 × Vaxzevria + 1 × Comirnaty RBD and sVNT Abs, respectively. (All data refer to measurements acquired using the Mindray method.)

**Table 1 biomedicines-11-02661-t001:** Demographics and vaccine type.

		Vaxzevria	Comirnaty
Gender	male	20	37
	female	36	76
Age (years)	<30	2	5
	30–50	20	68
	>50	34	40
Total		56	113

**Table 2 biomedicines-11-02661-t002:** Median and (1st–3rd) quartile data, calculated for Ab data obtained for both vaccine series, with both experimental setup versus vaccine dose (I/II and III).

Anti-RBD								
	3 × Comirnaty				2 × Vaxzevria + 1 × Comirnaty			
VACCINE DOSE	Mindray		Snibe		Mindray		Snibe	
	median	1st–3rd quartile	median	1st–3rd quartile	median	1st–3rd quartile	median	1st–3rd quartile
I/II	323	30–1504	359	45–1461	74	38–159	47	24–93
III	378	152–2592	219	89–4253	282	102–2436	1976	1062–6188
Anti-NT								
	3 × Comirnaty				2 × Vaxzevria + 1 × Comirnaty			
VACCINE DOSE	Mindray		Snibe		Mindray		Snibe	
	median	1st–3rd quartile	median	1st–3rd quartile	median	1st–3rd quartile	median	1st–3rd quartile
I/II	126	6–940	659	211–2058	14	6–35	96	59–160
III	465	189–4716	788	261–9667	151	32–933	7784	3100–11,850

## Data Availability

Not applicable.

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
