# Peer review of "The Antibodies’ Response to SARS-CoV-2 Vaccination: 1-Year Follow Up"

_biomedicines, 2023, doi:10.3390/biomedicines11102661_

Round 1

Reviewer 1 Report

The authors have conducted a small scale prospective measurement of IgG antibodies to the Receptor Binding Domain of the SARS-CoV-2 Spike protein post vaccination with Vaxrevria and Comirnaty (mRNA) COVID-19 vaccines using three different automated methods. As expected high antibody levels were  observed after completion of the second dose of each of the vaccines, but gradually declined over several months. Upon administering a third booster dose the antibody levels were significantly raised  leading the authors to conclude that both vaccines can induce long lasting protective immunity. However, it is known that the omicron variant of COVID-19 is not fully neutralised by vaccines raised against earlier variants of SARS-CoV-2. This limitation should be discussed by the authors and the conclusion appropriately nuance and shortened to 2-3 lines. Barring a few sentence fragments (line 382 and 392-395) the paper is well written and should be accepted for publication in MDPI after language editing.

See above

Author Response

The authors have conducted a small scale prospective measurement of IgG antibodies to the Receptor Binding Domain of the SARS-CoV-2 Spike protein post vaccination with Vaxrevria and Comirnaty (mRNA) COVID-19 vaccines using three different automated methods. As expected high antibody levels were observed after completion of the second dose of each of the vaccines, but gradually declined over several months. Upon administering a third booster dose the antibody levels were significantly raised leading the authors to conclude that both vaccines can induce long lasting protective immunity. However, it is known that the omicron variant of COVID-19 is not fully neutralised by vaccines raised against earlier variants of SARS-CoV-2. This limitation should be discussed by the authors and the conclusion appropriately nuance and shortened to 2-3 lines. Barring a few sentence fragments (line 382 and 392-395) the paper is well written and should be accepted for publication in MDPI after language editing.

We agree with the reviewer that the present vaccines don’t protect completely against infection by Omicron variants. Indeed, this point is addressed in Discussion, now lines 650-659, and Conclusions, lines 710-713. Furthermore, the sentence fragments (ex Line 382 and 392-395) have been modified as suggested by the reviewer.

Reviewer 2 Report

The study of Nicolai et has the potential to add a valuable piece to the puzzle of how well the global COVID vaccine campaign worked and aims to address a major concern that was raised after initial serological data from individuals who had received mRNA-based vaccines, ie that immunity might be waning too fast. The study is very straight forward, focusing on just two cohorts of vaccinees and measuring their antibody titers over time, as well as the antibodies’ ‘neutralizing’ activity based on a surrogate competition assay. 

-          One confusing aspect of the study is the immunization regimen: while the authors mention (in the abstract and line 101) that all participants received the mRNA vaccine as their final boost, they also talk about a third dose of the Vaxzevria vaccine (line 315). In general, adding a chart (timeline) that shows she immunization regimens for the two cohorts and that indicates when individuals were bled would be highly advisable.

-          As the authors point out in the discussion, the emergence of viral variants is a major confounding issue when talking about vaccine efficacy. Therefore, the authors need to make it clear which variants they are referring to when citing antibody titers of 200-800 BAU/ml as “protective”. In line 30, when claiming that “our data show a low risk of infection….”, the authors do not take into account the viral variants that were circulating at the time of the study. This conclusion would likely only apply to specific variants (and maybe only the ancestral virus).

-          In the description of the competition assay, the authors did not indicate which variant (presumably the ancestral/Wuhan line) the RBD reagent is based on.

-          Line 64: the authors claim that neutralizing antibody titers are strongly correlated with protection from infection (and did the studies they have in mind really report protection from infection, or rather protection from (severe) disease and death?), but then acknowledge that this claim has been disputed by other studies. If there is no reliable surrogate marker of vaccine efficacy, this sentence needs to be rephrased!

-          The authors used testing platforms for Ig titers and ‘neutralization’ from two different vendors. It is unclear why it was necessary to duplicate these readouts unless the authors also intended to formally compare the two platforms and test their reliability (which should have been spelled out and eg., a correlation of the date with the same samples should be provided).  Simply re-analysing the samples with a similar testing platform does not add any value to the study.

-          The data shown in Fig 1-3 are confusing for multiple reasons: 1) figure labels: instead of “II dose”, it should be something like “post 2nd dose”, 2) axis label: instead of “time (days)”, it should be something like “days after first immunization”. 3) collection points: In lines 102-108, the authors describe 6 serum collection time points that do not seem to match what is shown in the figures (time of immunization should be indicated on the X-axis). Based on how the data points are plotted, it appears that individuals were bled on very different days. 4) Vastly different data points are shown for the different sampling time points/time periods. Does this mean that different volunteers were bled for the different time points, or are eg., the 5 samples from the last time point(s?) for Comirnaty also consistently included in all the other time points? This inconsistency in the number of samples is a major flaw of the study – as the authors point out themselves, there is significant variability in the responses (as would be expected in a human cohort), so collecting samples from different donors at different time points would make it impossible to draw any conclusions about the time course of individual’s immune responses.

-          Line 215: what does “2522 antibody dosages” were tested mean? Are these antibody “samples”, and if all 113+56 volunteers had been bled 6 times as described (which they clearly were not since only a few data points are shown for certain time points), this would be a total (maximum) of 1014 samples.

-          Line 108: the authors describe having excluded volunteers from further participation when they tested positive for SARS-CoV-2 infection. This statement leaves a couple of questions open: when/how often did they test? What happened with the sera from these individuals collected at earlier time points – were they included up to the time they tested positive or where all of these samples excluded from the study? Including the earlier samples would skew the analysis of later time points since presumably those with higher antibody titers would have remained protected, and those data points would need to be identified (eg., different colour in the figure). Also, if the authors already had data on how many participants had COVID, why did they not include those here (and correlate with the volunteer’s antibody and NT titers) which would have significantly increased the impact of the study?

-          Figures 1-3: the authors are showing total Ig titers and NT titers separately – why was no attempt made to correlate the two (to support the hypothesis that they are correlated and to determine which vaccine/vaccine regimen is better at inducing not just antibodies, but ‘neutralization’).

-          Figure 4: The figure shows the time course of ab-titers for 5 “random” study participants – based on figures 1-3, the authors only had samples from 5 donors at some time points/time periods, so how can the selection of 5 donors be “random”? Also, this figure clearly indicates that donors were bled at different time points, again contradicting the sampling protocol described in lines 102-108.

-          Line 326: based on the data presented here, the authors cannot conclude that “the third dose can significantly increase the level of protection” since this conclusion is simply based on antibody titers and on other reports that showed protective efficacy of a certain titer against the ancestral strain (and, to some degree, against earlier variants such as Delta). However, efficacy against Omicron variants significantly dropped in individuals that received Wuhan S protein-based vaccines!

-          Line 398: The authors claim that IgG levels are a proxy for neutralizing antibody titers – that’s not correct. Whether or not antibody titers correlate with neutralization (and/or efficacy) depends on additional factors, such as the specific vaccine or the age of the vaccine recipient. The authors missed an opportunity to correlate those parameters in their own study. In general, it is also not appropriate to refer to the assay they used as a ‘neutralization assay’ – it is a competitive inhibition assay and does not involve actual virus (or pseudovirus) inhibition, so it’s essential that the authors acknowledge that they used a surrogate assay of neutralization, unless they include data or refer to studies (that were based on the specific systems they used) that correlate this inhibition assay with actual neutralization. The authors actually cite a paper (ref44) that addresses the differences between “neutralization assays”, but curiously they don’t refer to the relevant findings of that study to explain the usefulness of the assay they chose.

-          Line 236/37: assuming that the volunteers who initially received the ChimpAd vaccine were boosted with the mRNA vaccine, it is incorrect to assign the antibody titers from the Vaxzevria group at later time points to just one vaccine. Instead, throughout the manuscript, the authors need to clarify the regimen (3x mRNA vs. 2x Ad/1x mRNA). Similarly, it is inappropriate to label the figures “Comirnaty” and “Vaxzevria” since the latter group received a heterologous prime-boost-regimen.

-          Line 323: this sentence seems random and out of place- should be moved elsewhere. It also highlights another shortcoming of the study – the authors had collected certain, valuable information about the volunteers (namely age and sex) but did not include those in their analysis (ie., correlate antibody responses with those parameters).

-          The authors cite several papers that address the durability (only up to 6 months) of antibody responses after vaccination, including at least one looking at the same heterologous prime/boost regimen as the one described here, however, they don’t directly correlate those results with what they are seeing up to the 6month time-point. This would be very helpful considering that the authors used serology platforms that as common as those used in these other studies.

Minor points:

-          Throughout the manuscript, there are words with random hyphens (eg., line 52 “pro-duces”) that need to be removed.

-          Line 17: need to use the acronym (COVID-19), too, in order to help readers find the article.

-          Line 17: the issue was less the availability of vaccines, but their rollout and acceptance (once manufacturing had caught up with the initial demand, a lot of stockpiled vaccine ended up either not going to the right target population, eg., in developing countries, or was not accepted by a large part of the population)

-          Lines 22 and 23: since many readers won’t be familiar with the brand names, it would be helpful to add something like “mRNA-based” and “ChimpAd-based vaccine” to the names.

-          Line 30: the term “complete vaccination cycle” is problematic (also, should be “series”, not “cycle”) and should be avoided since there is not defined number of boosters an individual can/should get (comparable to the flu vaccine, and very different from eg. the shingles vaccine). In line 50, the authors suggest that Cormirnaty is “given in two doses 21d apart”. Again, this just happens to be the initial regimen (followed by a possibly infinite number of boosters), so it’s important to point out that this is the ”INITIAL vaccination regimen”.

-          Line 36: “tens of millions” is quite an understatement – this affected hundreds of millions of people, if not billions!

-          Line 44: “over 11 approved COVID vaccines” is also an understatement – more than 40 have received an EUA or BLA.

-          Line 44: ‘BC CENTRE’ is a local (British Columbia), Canadian health agency – why do the authors refer to it for a vaccine used in Italy?

-          Line 44: the authors picked Dec 14th 2020 as a key date for COVID vaccines because of the first immunization in the US – why is this date/event highlighted? The authors should also clarify whether they talk about the first immunization as part of a clinical trial, or the first recipient of a licensed vaccine. For example, the clinical trial of Covaxin started in June of 2020 already.

-          Line 52: what does “achieving high immunity” mean? This needs to be reworded or removed.

-          Line 54: what does “a second dose enhances the antibody levels, which are related to viral neutralization” mean? This needs to be rephrased.

-          Line 56: “antibody supply is scarce” suggests supply issues with recombinantly produced therapeutic antibodies but this sentence needs to be rephrased when referring to the low level of immunogenicity induced by a single dose of the vaccine. Also, the authors present the fact that the initial immunization of previously naïve individuals with a non-replicating vaccine only induces weak immunity as if it were surprising but virtually all non-replicating vaccines require at least 2 doses when given to naïve/seronegative individuals.

-          Line 58: this sentence needs to be rephrased/restructured!

-          Line 62: instead of “authorizations”, the authors presumably mean “regulatory authorities”?

-          Line 58: when describing and comparing the two vaccines used in this study, the authors need to actually compare them, namely the insert – only in case of the mRNA vaccine, they describe the insert (‘full-length S protein”), while in case of the ChimpAd vaccine, they only mention that the vector carries “the Spike protein”. Since the immunogenicity of the two vaccines is being directly compared, it’s important for readers to understand differences between the two inserts.

-          Line 66: mentioning humoral immunity again in this sentence is redundant since it was already discussed in the previous sentence.

-          Line 68: the term “patients” should be reserved for individuals who are/were infected, not vaccine recipients (vaccinees) who (as the authors pointed out) had not previously been infected. This sentence also needs to be rephrased.

-          Line 77: First, the term “fully vaccinated” (or “complete vaccination”, Line 91) is no longer valid or useful (as it was at the beginning of the pandemic). Instead, the number of doses that the vaccinees received should be spelled out. Second, “seropositive” is likely not the right term here – the authors seem to be referring to individuals with prior SARS-CoV-2 exposure (seropositivity can also be a result of vaccination, unless they only refer to anti-NP antibodies).

-          Line 86: It is unclear what the authors mean with the sentence starting with “A monitoring plan…”

-          Line 89: should be “recent studies”, not “works”

-          Line 96: what does “in health status” mean? Is this supposed to be “healthy adults”?

-          Line 108: should be SARS, not Sar

-          Line 109: “The positive outcomes” should be replaced with something like “positive cases” or “positive subjects”

-          Fig. 1: the red bar in panel b is outside the figure

-          Line 223: What does “After the second dose has been registered an increase…” mean? This sentence needs to be rephrased.

-          Figure legend 2: Should be “Neutralizing antibody levels”, not “Anti-NT levels”. Also, what does “considering the time interval” mean? This needs to be rephrased.

-          Line 320: “after one month” needs to be clarified – one month after the first or the second immunization?

-          Line 342: “it could be also interesting a comparative evaluation” is missing the verb (“…to conduct a…”)

-          Line 346: “triple dose vaccinated individuals” is the wrong term and suggests that they received 3-times the amount of vaccine for each shot!!

-          Line 347: as outlined above, the authors are not clearly describing the immunization regimen and assuming that the statement that everyone received an mRNA boost, this sentence is incorrect – Vaxzevria vaccine recipients had a lower response to that vaccine, but ended up with comparable titers to the Comirnaty group simply because they were boosted with the mRNA vaccine!!

-          Line 350/351: should be “in this case, Vaxzevria”, not “as Vaxzevria”

-          Line 387: authors suggest development of diagnostic methods for measuring secretory IgA – those already exist! The issue is implementation, not development of the tools. Overall, the discussion of IgA responses is irrelevant for this manuscript since no IgA was measured.

-          Line 397: should be “combating”, not “combacting”

-          Line 438: using the FDA’s postal address as a reference for the approval of a vaccine is not helpful – instead, the authors should cite the approval document (like they did for the EMA/ref 4)

The manuscript needs to be proof-read (and partially re-written) by a native English speaker. Under “minor points”, there are a couple of examples of the issues with wording and grammar that need to be addressed.

Author Response

The study of Nicolai et al. has the potential to add a valuable piece to the puzzle of how well the global COVID vaccine campaign worked and aims to address a major concern that was raised after initial serological data from individuals who had received mRNA-based vaccines, ie that immunity might be waning too fast. The study is very straight forward, focusing on just two cohorts of vaccinees and measuring their antibody titers over time, as well as the antibodies’ ‘neutralizing’ activity based on a surrogate competition assay. 

-          One confusing aspect of the study is the immunization regimen: while the authors mention (in the abstract and line 101) that all participants received the mRNA vaccine as their final boost, they also talk about a third dose of the Vaxzevria vaccine (line 315). In general, adding a chart (timeline) that shows she immunization regimens for the two cohorts and that indicates when individuals were bled would be highly advisable.

      We thank the reviewer for highlighting the misleading sentence at ex line 315. We corrected the sentence clarifying the meaning. Concerning the bled time, it is indicated in the graph (Figure 1) that has been modified accordingly to the reviewer suggestions, resulting now clearer.

-          As the authors point out in the discussion, the emergence of viral variants is a major confounding issue when talking about vaccine efficacy. Therefore, the authors need to make it clear which variants they are referring to when citing antibody titers of 200-800 BAU/ml as “protective”. In line 30, when claiming that “our data show a low risk of infection….”, the authors do not take into account the viral variants that were circulating at the time of the study. This conclusion would likely only apply to specific variants (and maybe only the ancestral virus).

      The sentence reported in the Abstract, ex line 30, is a general statement extrapolated from the literature and applies to those viral variants circulating in those countries at the time the studies were performed. In our case, during the study period considered (Juanuary 2021- February 2022) several viral variants circulated in Italy: B.1.1.7; B.1.617.2, and Omicron variants BA1 and BA2. Although, the current vaccines do not protect completely from these variants, a low risk of infection was observed in the population studied and the antibody titers measured fell within the range reported in literature and considered “protective”.

-          In the description of the competition assay, the authors did not indicate which variant (presumably the ancestral/Wuhan line) the RBD reagent is based on.

Both SNIBE and Mindray use the RBD antigen from the Wuhan strain. Thus, we think that also these assays may not fully reflect the real state of protection of an individual, as reported in discussion.

-          Line 64: the authors claim that neutralizing antibody titers are strongly correlated with protection from infection (and did the studies they have in mind really report protection from infection, or rather protection from (severe) disease and death?), but then acknowledge that this claim has been disputed by other studies. If there is no reliable surrogate marker of vaccine efficacy, this sentence needs to be rephrased!

We thank the reviewer for his/her objection. We modified the sentence by replacing the word “infection” with the word “disease”.

-          The authors used testing platforms for Ig titers and ‘neutralization’ from two different vendors. It is unclear why it was necessary to duplicate these readouts unless the authors also intended to formally compare the two platforms and test their reliability (which should have been spelled out and eg., a correlation of the date with the same samples should be provided).  Simply re-analysing the samples with a similar testing platform does not add any value to the study.

At the time of the surveillance, we couldn’t always rely on the availability of kits for the determination of antibodies against SARS-CoV-2 from only one vendor. For this reason, some samples were analyzed by Mindray and some by SNIBE, utilizing the same platform for the same individual at the different time points.

-          The data shown in Fig 1-3 are confusing for multiple reasons: 1) figure labels: instead of “II dose”, it should be something like “post 2nd dose”, 2) axis label: instead of “time (days)”, it should be something like “days after first immunization”. 3) collection points: In lines 102-108, the authors describe 6 serum collection time points that do not seem to match what is shown in the figures (time of immunization should be indicated on the X-axis). Based on how the data points are plotted, it appears that individuals were bled on very different days. 4) Vastly different data points are shown for the different sampling time points/time periods. Does this mean that different volunteers were bled for the different time points, or are eg., the 5 samples from the last time point(s?) for Comirnaty also consistently included in all the other time points? This inconsistency in the number of samples is a major flaw of the study – as the authors point out themselves, there is significant variability in the responses (as would be expected in a human cohort), so collecting samples from different donors at different time points would make it impossible to draw any conclusions about the time course of individual’s immune responses.

  1. the label “II DOSE” indicates the day of the second vaccine injection. To make it clearer, we added 2 dotted lines corresponding to the immunization times.
  2. corrected
  3. The samples collection time schedule reported in line 102-108, represent roughly the times of the collection, since not all volunteers were available the same day. Indeed, we modified the text inserting the word “about” before the days.
  4. As mentioned in point 3, the bled time points were not the same for all volunteers. The graph shows all the antibody titers data collected.

-          Line 215: what does “2522 antibody dosages” were tested mean? Are these antibody “samples”, and if all 113+56 volunteers had been bled 6 times as described (which they clearly were not since only a few data points are shown for certain time points), this would be a total (maximum) of 1014 samples.

The total number of dosages came from the total samples collected, and analyzed for anti S-RBD and NT antibodies.

-          Line 108: the authors describe having excluded volunteers from further participation when they tested positive for SARS-CoV-2 infection. This statement leaves a couple of questions open: when/how often did they test? What happened with the sera from these individuals collected at earlier time points – were they included up to the time they tested positive or where all of these samples excluded from the study? Including the earlier samples would skew the analysis of later time points since presumably those with higher antibody titers would have remained protected, and those data points would need to be identified (eg., different colour in the figure). Also, if the authors already had data on how many participants had COVID, why did they not include those here (and correlate with the volunteer’s antibody and NT titers) which would have significantly increased the impact of the study?

 Our cohort was monitored by RT-PCR assay performed on nasopharyngeal swab. The study was conducted on healthcare and University personnel, who were periodically monitored for SARS-CoV-2 by nasopharyngeal swab. Individuals who tested positive on the swab were excluded from the study. Indeed, the individuals who tested positive for SARS-CoV-2 infection were found to be 4. They were excluded from the study as such a small sample size would not have allowed for any statistical analysis. No data of these individuals were reported in the graphs. Additionally, the individuals who tested positive were no longer following the vaccination schedule. However, our group has already studied in a previous work the synergistic effect of vaccine and infection on the antibody production. (ref.: 51)

-          Figures 1-3: the authors are showing total Ig titers and NT titers separately – why was no attempt made to correlate the two (to support the hypothesis that they are correlated and to determine which vaccine/vaccine regimen is better at inducing not just antibodies, but ‘neutralization’).

RBD and NT titers have different unit measurements, for this they cannot be compared directly.

-          Figure 4: The figure shows the time course of ab-titers for 5 “random” study participants – based on figures 1-3, the authors only had samples from 5 donors at some time points/time periods, so how can the selection of 5 donors be “random”? Also, this figure clearly indicates that donors were bled at different time points, again contradicting the sampling protocol described in lines 102-108.

Figures 1-3 report the trend over time of data obtained from all donors samples. For the sake of clarity, the data are reported as a scatter plot with no connection between points. Figure 4 indeed, shows 5 of these trends, to highlight their individuality.

-          Line 326: based on the data presented here, the authors cannot conclude that “the third dose can significantly increase the level of protection” since this conclusion is simply based on antibody titers and on other reports that showed protective efficacy of a certain titer against the ancestral strain (and, to some degree, against earlier variants such as Delta). However, efficacy against Omicron variants significantly dropped in individuals that received Wuhan S protein-based vaccines!

It is true that protection against Omicron variants is not very effective with the current vaccines and that the antibody titers decline over time, but the third booster dose increased again the titer besides recalling memory T cells.

-          Line 398: The authors claim that IgG levels are a proxy for neutralizing antibody titers – that’s not correct. Whether or not antibody titers correlate with neutralization (and/or efficacy) depends on additional factors, such as the specific vaccine or the age of the vaccine recipient. The authors missed an opportunity to correlate those parameters in their own study.

We monitored people who received two vaccines, and we report and discuss the difference between them, in term of Abs titers; we studied a possible relationship with age, but we didn’t find any correlation, probably due to the fact that our cohort is quite homogeneous in terms of age.

In general, it is also not appropriate to refer to the assay they used as a ‘neutralization assay’ – it is a competitive inhibition assay and does not involve actual virus (or pseudovirus) inhibition, so it’s essential that the authors acknowledge that they used a surrogate assay of neutralization, unless they include data or refer to studies (that were based on the specific systems they used) that correlate this inhibition assay with actual neutralization. The authors actually cite a paper (ref44) that addresses the differences between “neutralization assays”, but curiously they don’t refer to the relevant findings of that study to explain the usefulness of the assay they chose.

We agree with the reviewer, it is incorrect to call them "neutralization tests" and for this reason we have decided to call them Surrogate Virus Neutralization Tests (sVNT). However, I want to point out to the reviewer that these tests have shown, although virus neutralization test (VNT) still remains the gold standard, a good agreement between VNT and sVNT equal to 98.86 (%) as described in our previous paper by Pieri et al. in front of. Biosci. (Landmark Ed) 2022; 27(2): 074.

We have thus modified Neutralization Tests with surrogate Virus Neutralization Tests (sVNTs) and inserted a sentence in the discussion with the bibliographic reference.

-          Line 236/37: assuming that the volunteers who initially received the ChimpAd vaccine were boosted with the mRNA vaccine, it is incorrect to assign the antibody titers from the Vaxzevria group at later time points to just one vaccine. Instead, throughout the manuscript, the authors need to clarify the regimen (3x mRNA vs. 2x Ad/1x mRNA). Similarly, it is inappropriate to label the figures “Comirnaty” and “Vaxzevria” since the latter group received a heterologous prime-boost-regimen.

Thank you to the reviewer for his/her observation we re-drawn the figures modifying the labels and we also modified the definitions in the text.

-          Line 323: this sentence seems random and out of place- should be moved elsewhere. It also highlights another shortcoming of the study – the authors had collected certain, valuable information about the volunteers (namely age and sex) but did not include those in their analysis (ie., correlate antibody responses with those parameters).

Thank you to the reviewer for his/her observation. Indeed the sentence wants to highlight the interpersonal variability between different people, speculating on the individuality in the Abs response.

-          The authors cite several papers that address the durability (only up to 6 months) of antibody responses after vaccination, including at least one looking at the same heterologous prime/boost regimen as the one described here, however, they don’t directly correlate those results with what they are seeing up to the 6month time-point. This would be very helpful considering that the authors used serology platforms that as common as those used in these other studies.

A sentence has been added in the discussion

Minor points:

-          Throughout the manuscript, there are words with random hyphens (eg., line 52 “pro-duces”) that need to be removed.

Wrong hyphens have been removed

-          Line 17: need to use the acronym (COVID-19), too, in order to help readers find the article.

Thank you for the suggestion, the acronym has been added.

-          Line 17: the issue was less the availability of vaccines, but their rollout and acceptance (once manufacturing had caught up with the initial demand, a lot of stockpiled vaccine ended up either not going to the right target population, eg., in developing countries, or was not accepted by a large part of the population)

The word “availability” has been changed in “use”

-          Lines 22 and 23: since many readers won’t be familiar with the brand names, it would be helpful to add something like “mRNA-based” and “ChimpAd-based vaccine” to the names.

To not exceed the word limit allowed in the abstract, the specification of the vaccines have been reported in the introduction.

-          Line 30: the term “complete vaccination cycle” is problematic (also, should be “series”, not “cycle”) and should be avoided since there is not defined number of boosters an individual can/should get (comparable to the flu vaccine, and very different from eg. the shingles vaccine).

We rephrased the sentence according to the reviewer suggestion

  • In line 50, the authors suggest that Cormirnaty is “given in two doses 21d apart”. Again, this just happens to be the initial regimen (followed by a possibly infinite number of boosters), so it’s important to point out that this is the ”INITIALvaccination regimen”.

Thank you to the reviewer, the specification has been inserted in the text.

-          Line 36: “tens of millions” is quite an understatement – this affected hundreds of millions of people, if not billions!

The sentence has been modified

-          Line 44: “over 11 approved COVID vaccines” is also an understatement – more than 40 have received an EUA or BLA.

Thank you for the observation, we modified the text accordingly.

-          Line 44: ‘BC CENTRE’ is a local (British Columbia), Canadian health agency – why do the authors refer to it for a vaccine used in Italy?

We mentioned the BC CENTRE because at the time of writing we collected the information form the website of that agency. 

-          Line 44: the authors picked Dec 14th 2020 as a key date for COVID vaccines because of the first immunization in the US – why is this date/event highlighted? The authors should also clarify whether they talk about the first immunization as part of a clinical trial, or the first recipient of a licensed vaccine. For example, the clinical trial of Covaxin started in June of 2020 already.

The date of Dec 14 was highlighted as a starting date of the vaccination campaign.

-          Line 52: what does “achieving high immunity” mean? This needs to be reworded or removed.

High immunity refers to the 95% protection against COVID-19 observed after a two-dose regimen of BNT162b2, as stated in the reference cited.          

Line 54: what does “a second dose enhances the antibody levels, which are related to viral neutralization” mean? This needs to be rephrased.

The sentence has been rephrased as suggested.

-          Line 56: “antibody supply is scarce” suggests supply issues with recombinantly produced therapeutic antibodies but this sentence needs to be rephrased when referring to the low level of immunogenicity induced by a single dose of the vaccine. Also, the authors present the fact that the initial immunization of previously naïve individuals with a non-replicating vaccine only induces weak immunity as if it were surprising but virtually all non-replicating vaccines require at least 2 doses when given to naïve/seronegative individuals.

The sentence has been rephrased as suggested.

-          Line 58: this sentence needs to be rephrased/restructured!

The sentence in line 58: “Vaxzevria is an adenovirus-based vaccine, replication deficient, which has been modified to contain the gene encoding the Spike protein of SARS-CoV-2 [11]”, it looks correct to us. It is a commonly used definition.

-          Line 62: instead of “authorizations”, the authors presumably mean “regulatory authorities”?

The sentence has been corrected.

-          Line 58: when describing and comparing the two vaccines used in this study, the authors need to actually compare them, namely the insert – only in case of the mRNA vaccine, they describe the insert (‘full-length S protein”), while in case of the ChimpAd vaccine, they only mention that the vector carries “the Spike protein”. Since the immunogenicity of the two vaccines is being directly compared, it’s important for readers to understand differences between the two inserts.

Sentence in line 58 fully describes the design of the vector encoding for the Spike protein of the SARS -CoV-2. While about is mentioned that the Comirnaty vaccine is made by a mRNA encoding for the Spike protein of SARS-CoV-2.

-          Line 66: mentioning humoral immunity again in this sentence is redundant since it was already discussed in the previous sentence.

This is the first time we mention “humoral immunity”, it is not present in the previous sentence as indicated by the reviewer.

-          Line 68: the term “patients” should be reserved for individuals who are/were infected, not vaccine recipients (vaccinees) who (as the authors pointed out) had not previously been infected. This sentence also needs to be rephrased.

The sentence has been rephrased

-          Line 77: First, the term “fully vaccinated” (or “complete vaccination”, Line 91) is no longer valid or useful (as it was at the beginning of the pandemic). Instead, the number of doses that the vaccinees received should be spelled out. Second, “seropositive” is likely not the right term here – the authors seem to be referring to individuals with prior SARS-CoV-2 exposure (seropositivity can also be a result of vaccination, unless they only refer to anti-NP antibodies).

The sentence has been changed

-          Line 86: It is unclear what the authors mean with the sentence starting with “A monitoring plan…”

The sentence has been changed

-          Line 89: should be “recent studies”, not “works”

Corrected

-          Line 96: what does “in health status” mean? Is this supposed to be “healthy adults”?

Thank you to the reviewer for the correction

-          Line 108: should be SARS, not Sar

Thank you to the reviewer for the correction

-          Line 109: “The positive outcomes” should be replaced with something like “positive cases” or “positive subjects”

corrected

-          Fig. 1: the red bar in panel b is outside the figure

corrected

-          Line 223: What does “After the second dose has been registered an increase…” mean? This sentence needs to be rephrased.

The sentence has been rephrased

-          Figure legend 2: Should be “Neutralizing antibody levels”, not “Anti-NT levels”.

We made the correction

Also, what does “considering the time interval” mean? This needs to be rephrased.

The statement has been rephrased

-          Line 320: “after one month” needs to be clarified – one month after the first or the second immunization?

The sentence has been changed

-          Line 342: “it could be also interesting a comparative evaluation” is missing the verb (“…to conduct a…”)

Thank you for the correction

-          Line 346: “triple dose vaccinated individuals” is the wrong term and suggests that they received 3-times the amount of vaccine for each shot!!

The sentence has been modified according to the reviewer suggestion.

-          Line 347: as outlined above, the authors are not clearly describing the immunization regimen and assuming that the statement that everyone received an mRNA boost, this sentence is incorrect – Vaxzevria vaccine recipients had a lower response to that vaccine, but ended up with comparable titers to the Comirnaty group simply because they were boosted with the mRNA vaccine!!

The sentence has been modified

-          Line 350/351: should be “in this case, Vaxzevria”, not “as Vaxzevria”

Thank you to the reviewer for the correction

-          Line 387: authors suggest development of diagnostic methods for measuring secretory IgA – those already exist! The issue is implementation, not development of the tools.

We made the correction

Overall, the discussion of IgA responses is irrelevant for this manuscript since no IgA was measured.

Even though we haven’t measured IgA in this study, we introduce this consideration at the end of the discussion because the development of mucosal vaccines could be important for the future containment of the SARS-CoV-2 spread.  

-          Line 397: should be “combating”, not “combacting”

We made the correction

-          Line 438: using the FDA’s postal address as a reference for the approval of a vaccine is not helpful – instead, the authors should cite the approval document (like they did for the EMA/ref 4)

The reference has been corrected.

The manuscript needs to be proof-read (and partially re-written) by a native English speaker. Under “minor points”, there are a couple of examples of the issues with wording and grammar that need to be addressed.

Thank you to the reviewer for his/her corrections and suggestions. We addressed the highlighted points and revised english throughtout the manuscript.

Round 2

Reviewer 2 Report

The authors were quite responsive to comments and concerns from the first round of reviews and have significantly improved the readability of the manuscript. However, a number of issues persists, including the need to have the manuscript proof-read by a native English speaker.

-          A question in the previous round was about the statement that the authors analyzed “2522 dosages”. First, the term “dosage” is wrong in this context and still has not been corrected – should be “samples”. Second, the authors still haven’t addressed the question how they came up with that number considering that they refer to 168 volunteers and 6 bleeds, this number does not seem to make sense.

-          In response to the question why the authors did not include a simple graph that correlates RBD and NT-titers, the authors argued that since the two measures use different units, they cannot be correlated. That assumption is fundamentally wrong – any parameters can be correlated with each other!

-          In response to the question why the authors did not correlate age of donors and antibody titers, even though they repeatedly discuss this aspect, the authors argue that the age range of their donors was very tight. However, in Table 1, they show a wide range of ages. Please explain this discrepancy.

-          The previous review pointed out that in addition to not looking at age (the authors now argue that they found no correlation there), the manuscript also didn’t attempt to correlate other parameters (eg., sex) with vaccine-induced responses. The authors respond by emphasizing again that many parameters determine individual responses, but do not address the question – if no other correlations were found, why are the authors not including that finding in their discussion?

-          As previously pointed out, the terminology “triple dose” is highly misleading since it suggests that 3 times the normal dose was used for a single immunization. Instead, use something like “individuals who had been vaccinated three times” or “who had received two booster shots”

-          Line 389: the first half of this paragraph is confusing and somewhat out of place. The authors bring up the issue of ADE, a hypothetical problem that was discussed at the beginning of the pandemic, but has not panned out even after vaccination with alum-adjuvanted vaccines. This makes SARS-CoV-2 different from SARS-1 and it’s unclear why the authors include such a discussion. The reference (45) cited is also not useful – it’s for a comment of a publication on decaying antibody titers to SARS-CoV-2. The second half of the paragraph is much more useful (factors that affect individual’s antibody titers) and should be the only aspect of this paragraph.

-          Reference 3: while the authors removed the mailing address of the FDA, they are still just referring to the agency, rather than an actual document (such as the agency’s guidance document)

-          In the previous review, I pointed out that the statement about having used “5 random donors” for the time-course analysis did not seem to make sense since for some of the time points, the authors did not seem to have more than 5 samples for some of the time points. The authors did not address this concern.

-          The previous comment about the sentence in Line 58 refers to the wording: should be “Vaxzevria is a replication-deficient, adenovirus-based vaccine”

Minor points

-          Fig. 1 – the authors have added a line to indicate when the booster was received, which is very helpful, but the label should be changed from “II dose” to “2nd dose” or “first booster” (same for “III dose”)

-         

-          Line 90: as pointed out previously, the reference to individuals from a hospital is misleading since it suggests that this study focused on COVID patients, not healthy vaccine recipients

-          Line 325: to say “ the antibody trend as a function of time follows that shown in previous figures” is an inappropriate, circular argument since the data shown here are, after all, included in – and extracted from – those previous figures!!

-          Line 338: in other parts of the manuscript, the authors already addressed inappropriate referrals to protection, but missed this sentence – the results do NOT clearly show that the third dose can “increase the level of protection” since no protection is measured/reported, only antibody titers and a surrogate measure of neutralizing antibody activity! This needs to be rephrased.

-          Line 344: incomplete sentence – “and continue up to 8 months” should be “and continue to decrease up to…”

-          Line 350: “SARS-COV-2” should be “SARS-CoV-2”

-          Line 429: the authors refer to “the voluntary screening program”, but don’t specify which one they are referring to.

-          While the authors removed most of the rogue hyphens, some are still left – eg., in line 103 (“Uni-versity”), or Line 160 (“anti-bodies”)

-          Line 104: “while hospital workers” – is missing a verb

-          Line 179: a linearity “range” requires that a lower and higher cutoff value are being listed – the authors only cited one.

-          The authors occasionally refer to RBD as “S-RBD” without further explanation. It is unclear what this means – if this refers to a soluble, recombinant protein (used in the assay), the term cannot be used when referring to the RBD sequence within the S protein.

 As pointed out previously, the manuscript would greatly benefit from being proof-read by a native English speaker. The authors made some improvements to in terms of incorrect or imprecise grammar and improper use of words, but problems persist and the manuscript should STILL be proof read by a native English speaker. Examples:

o   Line 31: “undergoing to a three vaccine doses” needs to be completely rephrased (e.g., individuals who received three COVID vaccinations)

o   Line 43: instead of “genome characterization”, use something like “sequencing of its genome”

o   Line 45: “authorities” should be “regulatory authorities”

o   Throughout the manuscript: “the manufacturer declares” should be “the manufacturer reports”;

o   Line 209: “In the last two years” should be “Over the course of two years”

o   Line 219: “12 weeks apart the Vaxzevria, …” – should be “12 weeks apart FOR Vaxzevria, ….”

o   Line 224: “ Because of the second dose has been administered at different time…” should be “Because the second dose was administered at a different time point….”

o   Line 229: instead of “registered”, use another term such as “detected” or “measured”

o   Line 234: “individuals who received the Vaxzevria vaccine” should be “individuals who had initially received the Vaxzevria vaccine”

o   Lines 253/269/298 etc: replace “dose injection” with something like “booster immunization”

o   Lines 254/270: “ reported in literature considered protective” should be something like “reported in THE literature to be protective”

o   Line 277:  verb is missing (“and ARE reported in  Fig. 3”)

o   Line 321: should be “Antibody levels”, not “antibodies levels” (recommend searching the entire manuscript for remaining instances of this mistake)

o   Line 332: “on the contrary people” should be “in contrast to people”

o   Line 332: “inizially” should be “initially”

o   Line 354: “at the moment, contrasting data were reported by previous works” should be something like “Contradictory data have been reported by others”

o   Line 367: “Figure 4 where can be observed” should be something like “Figure 4 that shows…”

o   Line 369: the phrase “the booster does” can be removed since it’s misleading (the second dose is already a booster)

o   Line 372: should be “in previous studies” instead of “in previous findings”

o   Line 384: “today vaccines” should be something like “current COVID vaccines” (or, to be more specific, “the original COVID vaccines”)

o   Line 99: remove “people” (redundant, since you already call them “healthy individuals”

o   Line 107: “blood drown collection” needs to be replaced with something like “blood collections” (or “blood DRAWS”)

o   Line 115: “accordance to” should be “accordance with”

o   Line 125: sentence is missing a verb (“is”)

o   Line 224: “Because of the second dose has been administered” should be “Because the second dose had been administered”

Author Response

Dear reviewer, please find attached the answers to your comments. We hope to have addressed all the issues raised. The manuscript has been proof-read by a native English speaker. 

-          A question in the previous round was about the statement that the authors analyzed “2522 dosages”. First, the term “dosage” is wrong in this context and still has not been corrected – should be “samples”. Second, the authors still haven’t addressed the question how they came up with that number considering that they refer to 168 volunteers and 6 bleeds, this number does not seem to make sense.

The term “sample” would not be correct. From bleeds, we obtained generally 6 samples from each volunteer, but it could happen that some volunteers missed one bleed or others got more than 6. Samples obtained by bleeds were measured to get the concentration of RBD and sVNT antibodies, with 2 methods: Mindray and SNIBE. So, for each volunteer we got in most case 6 samples (but in same cases they were less or more), and the antibodies titer in each of these samples were dosed (hence the term “dosage”) a maximum of 4 times: samples dosed 4 times were dosed for RBD and sVNT by Mindray and SNIBE, but the number of dosages was dictated by the availability of the measurements kit.

-          In response to the question why the authors did not include a simple graph that correlates RBD and NT-titers, the authors argued that since the two measures use different units, they cannot be correlated. That assumption is fundamentally wrong – any parameters can be correlated with each other!

Indeed, figure 3 shows RBD behavior versus time next to sVNT behavior versus time, in order to highlight the analogous trend observed for both antibodies, as stated in the results. In the previous round answer we meant that RBD and svNT couldn’t be compared directly in the same graph, but we compared them in different but close graphs.

-          In response to the question why the authors did not correlate age of donors and antibody titers, even though they repeatedly discuss this aspect, the authors argue that the age range of their donors was very tight. However, in Table 1, they show a wide range of ages. Please explain this discrepancy.

As concern the correlation between age and antibody titer, we wrote that in literature has been found a correlation unlike us. We tried to make a statistical correlation, but it was not significant. A possible explanation could be that the majority of volunteers fell in the range 40-50 years old. Because of the lack of statistical significance, in the table we decided not to detail the other age groups.

-          The previous review pointed out that in addition to not looking at age (the authors now argue that they found no correlation there), the manuscript also didn’t attempt to correlate other parameters (eg., sex) with vaccine-induced responses. The authors respond by emphasizing again that many parameters determine individual responses, but do not address the question – if no other correlations were found, why are the authors not including that finding in their discussion?

We did not find any correlation between age or sex and antibodies titer, and we did not explore correlation with other factors (mentioned in literature) because we did not have such information.

-          As previously pointed out, the terminology “triple dose” is highly misleading since it suggests that 3 times the normal dose was used for a single immunization. Instead, use something like “individuals who had been vaccinated three times” or “who had received two booster shots”

      Indeed, we accepted your previous suggestion and removed this terminology from the text.

-          Line 389: the first half of this paragraph is confusing and somewhat out of place. The authors bring up the issue of ADE, a hypothetical problem that was discussed at the beginning of the pandemic, but has not panned out even after vaccination with alum-adjuvanted vaccines. This makes SARS-CoV-2 different from SARS-1 and it’s unclear why the authors include such a discussion. The reference (45) cited is also not useful – it’s for a comment of a publication on decaying antibody titers to SARS-CoV-2. The second half of the paragraph is much more useful (factors that affect individual’s antibody titers) and should be the only aspect of this paragraph.

According to your suggestion we deleted the sentence referred to ADE and the related reference.

-          Reference 3: while the authors removed the mailing address of the FDA, they are still just referring to the agency, rather than an actual document (such as the agency’s guidance document)

      We edited reference 3 indicating the document we referred to that can be downloaded at: https://www.regulations.gov/docket/FDA-2020-D-1137/document.

-          In the previous review, I pointed out that the statement about having used “5 random donors” for the time-course analysis did not seem to make sense since for some of the time points, the authors did not seem to have more than 5 samples for some of the time points. The authors did not address this concern.

We didn’t get the point of the reviewer. Figure 4 shows 5 antibody trend from 5 different volunteers. The data that make up the trend belong to different time points. For example, for panel a, the light blue line has points at 0, 38, 70, 100, 157, 210, 240, 310 days after first immunization; instead, the yellow line got points at: 0, 10, 35, 50, 75, 135, 225, 255, 310 days after first immunization. We didn’t report 5 antibodies data at the same bleed time.

-          The previous comment about the sentence in Line 58 refers to the wording: should be “Vaxzevria is a replication-deficient, adenovirus-based vaccine”

The sentence has been corrected

Minor points

-          Fig. 1 – the authors have added a line to indicate when the booster was received, which is very helpful, but the label should be changed from “II dose” to “2nd dose” or “first booster” (same for “III dose”)

The labels have been corrected

-          Line 90: as pointed out previously, the reference to individuals from a hospital is misleading since it suggests that this study focused on COVID patients, not healthy vaccine recipients

We corrected the sentence.

-          Line 325: to say “ the antibody trend as a function of time follows that shown in previous figures” is an inappropriate, circular argument since the data shown here are, after all, included in – and extracted from – those previous figures!!

The sentence has been corrected

-          Line 338 in other parts of the manuscript, the authors already addressed inappropriate referrals to protection, but missed this sentence – the results do NOT clearly show that the third dose can “increase the level of protection” since no protection is measured/reported, only antibody titers and a surrogate measure of neutralizing antibody activity! This needs to be rephrased.

The sentence has been rephrased.

-          Line 344: incomplete sentence – “and continue up to 8 months” should be “and continue to decrease up to…”

The sentence has been corrected

-          Line 350: “SARS-COV-2” should be “SARS-CoV-2”

It has been corrected

-          Line 429: the authors refer to “the voluntary screening program”, but don’t specify which one they are referring to.

      We are referring to the one conducted in our institution and reported in this study. To make it clearer, we rephrased the sentence.

-          While the authors removed most of the rogue hyphens, some are still left – eg., in line 103 (“Uni-versity”), or Line 160 (“anti-bodies”)

The two hyphens have been removed.

-          Line 104: “while hospital workers” – is missing a verb

Sentence corrected

-          Line 179: a linearity “range” requires that a lower and higher cutoff value are being listed – the authors only cited one.

Thank you to the reviewer for his/her comment, we corrected the sentence, including the linear range declared by manufacturer.

-          The authors occasionally refer to RBD as “S-RBD” without further explanation. It is unclear what this means – if this refers to a soluble, recombinant protein (used in the assay), the term cannot be used when referring to the RBD sequence within the S protein.

We corrected the terms S-RBD in RBD, throughout the text. We only left S-RBD when we refer to the name of the measurements kit used.

Comments on the Quality of English Language

 As pointed out previously, the manuscript would greatly benefit from being proof-read by a native English speaker. The authors made some improvements to in terms of incorrect or imprecise grammar and improper use of words, but problems persist and the manuscript should STILL be proof read by a native English speaker. Examples:

Line 31: “undergoing to a three vaccine doses” needs to be completely rephrased (e.g., individuals who received three COVID vaccinations)

corrected

Line 43: instead of “genome characterization”, use something like “sequencing of its genome”

corrected

Line 45: “authorities” should be “regulatory authorities”

corrected

Throughout the manuscript: “the manufacturer declares” should be “the manufacturer reports”;

corrected

Line 209: “In the last two years” should be “Over the course of two years”

corrected

Line 219: “12 weeks apart the Vaxzevria, …” – should be “12 weeks apart FOR Vaxzevria, ….”

corrected

Line 224: “ Because of the second dose has been administered at different time…” should be “Because the second dose was administered at a different time point….”

corrected

Line 229: instead of “registered”, use another term such as “detected” or “measured”

corrected

Line 234: “individuals who received the Vaxzevria vaccine” should be “individuals who had initially received the Vaxzevria vaccine”

corrected

Lines 253/269/298 etc: replace “dose injection” with something like “booster immunization”

corrected

Lines 254/270: “ reported in literature considered protective” should be something like “reported in THE literature to be protective”

corrected

Line 277:  verb is missing (“and ARE reported in  Fig. 3”)

corrected

Line 321: should be “Antibody levels”, not “antibodies levels” (recommend searching the entire manuscript for remaining instances of this mistake)

corrected

Line 332: “on the contrary people” should be “in contrast to people”

corrected

Line 332: “inizially” should be “initially” 

corrected

Line 354: “at the moment, contrasting data were reported by previous works” should be something like “Contradictory data have been reported by others”

corrected

 Line 367: “Figure 4 where can be observed” should be something like “Figure 4 that shows…” 

corrected

Line 369: the phrase “the booster does” can be removed since it’s misleading (the second dose is already a booster)

corrected

Line 372: should be “in previous studies” instead of “in previous findings” 

corrected

Line 384: “today vaccines” should be something like “current COVID vaccines” (or, to be more specific, “the original COVID vaccines”)

corrected

Line 99: remove “people” (redundant, since you already call them “healthy individuals” 

corrected

Line 107: “blood drown collection” needs to be replaced with something like “blood collections” (or “blood DRAWS”) 

corrected

Line 115: “accordance to” should be “accordance with”

corrected

Line 125: sentence is missing a verb (“is”) 

Corrected

Line 224: “Because of the second dose has been administered” should be “Because the second dose had been administered”  

corrected
